# Activation of fibroblast growth factor-inducible 14 in the early phase of childhood IgA nephropathy

Yuko Tezuka[1☯¤a], Minenori Eguchi-Ishimae[1☯], Erina Ozaki[2], Toshiyuki Ito[1¤b], Eiichi Ishii[1¤c], Mariko Eguchi[1,3]*

**1** Department of Pediatrics, Takamatsu Red Cross Hospital, Takamatsu, Kagawa, Japan, **2** Department of Total Medical Support Center, Ehime University Hospital, Toon, Ehime, Japan, **3** Division of Medical Genetics, Ehime University Hospital, Toon, Ehime, Japan

☯ These authors contributed equally to this work.
¤a Current address: Department of Pediatrics, Ehime Prefectural Niihama Hospital, Niihama, Ehime, Japan
¤b Current address: Department of Pediatrics, Uwajima City Hospital, Uwajima, Ehime, Japan
¤c Current address: Imabari City Medical Association Hospital, Imabari, Ehime, Japan
* maeguchi@m.ehime-u.ac.jp

**Data Availability Statement:** All relevant data except transcriptome data, are within the manuscript. Raw data of transcriptome analysis

## Abstract

IgA nephropathy (IgAN) is the most common form of glomerulonephritis worldwide. Pediatric patients in Japan are diagnosed with IgAN at an early stage of the disease through annual urinary examinations. Tumor necrosis factor-like weak inducer of apoptosis (TWEAK) and fibroblast growth factor-inducible 14 (Fn14) have various roles, including proinflammatory effects, and modulation of several kidney diseases; however, no reports have described their roles in pediatric IgAN. In this study, we performed pathological and immunohistochemical analyses of samples from 14 pediatric IgAN patients. Additionally, gene expression arrays of glomeruli by laser-captured microdissection were performed in hemi-nephrectomized high serum IgA (HIGA) mice, a model of IgA nephropathy, to determine the role of Fn14. Glomeruli with intense Fn14 deposition were observed in 80% of mild IgAN cases; however, most severe cases showed glomeruli with little or no Fn14 deposition. Fn14 deposition was not observed in obvious mesangial proliferation or the crescent region of glomeruli, but was detected strongly in the glomerular tuft, with an intact appearance. In HIGA mice, Fn14 deposition was observed mildly beginning at 11 weeks of age, and stronger Fn14 deposition was detected at 14 weeks of age. Expression array analysis indicated that Fn14 expression was higher in HIGA mice at 6 weeks of age, increased slightly at 11 weeks, and then decreased at 26 weeks when compared with controls at equivalent ages. These findings suggest that Fn14 signaling affects early lesions but not advanced lesions in patients with IgAN. Further study of the TWEAK/Fn14 pathway will contribute to our understanding of the progression of IgAN.

will be available at DDBJ Sequence Read Archive
(DRA).

**Funding:** The author(s) received no specific
funding for this work.

**Competing interests:** The authors have declared
that no competing interests exist.

## Introduction

IgA nephropathy (IgAN) is the most common form of glomerulonephritis worldwide and an
important cause of end-stage renal disease (ESRD), particularly in young adults [1, 2]. IgAN is
defined by the presence of IgA-dominant or co-dominant immune deposits within glomeruli,
as shown by immunohistology [3].

In Japan, all children between the ages of 6 and 18 years are screened annually by urinalysis
via a school screening program, facilitating the early diagnosis of IgAN through the detection
of asymptomatic hematuria [4, 5]. These annual urinary examinations enabled us to diagnose
IgAN within 1 year from the actual onset of nephropathy. The school screening program has
identified 73.6% of patients with IgAN showing minimal proteinuria ($< 0.5$ g/day/1.73 m$^2$),
whose mesangial proliferative lesions were significantly mild according to the Oxford criteria
M0 in 76% of cases [6].

The progression of IgAN to ESRD is slower in children than in adults, probably due to early
diagnosis [4, 5]. Many childhood cases will never show progression, sometimes resulting in
spontaneous remission in cases of mild IgAN. Some pediatric patients progress to ESRD dur-
ing childhood, however, and many have slow yet continuous progression during adulthood
[6]. Among pediatric patients with IgA nephropathy in Japan, 11% have been reported to
develop ESRD within 15 years [7]. It was reported that the clinical predictors of a poor out-
come were a low glomerular filtration rate (GFR), a high mean blood pressure and a high
amount of albuminuria at time of biopsy, and low GFR and a high albuminuria during follow
up. Further histological predictors included mesangial hypercellularity, endocapillary hyper-
cellularity, tubular atrophy and crescents [8]. While the incidence of pediatric IgA nephropa-
thy patients who show hypertension or decreased renal function at onset is relatively rare in
comparison with adults, proteinuria is the most important risk factor for progression of the
disease in childhood, in particular, the degree of proteinuria during the follow-up period [9]. It
is difficult to predict outcomes at disease onset. In this regard, identification of specific marker
(s) that indicate the necessity for therapeutic intervention would be useful to determine treat-
ment and follow-up strategies.

In pediatric IgAN, glomerular lesions are characterized by increased numbers of mesangial
cells accompanied by a mesangial extracellular matrix. These pathological changes are thought
to represent early IgAN lesions in young patients [10].

Binding of tumor necrosis factor (TNF) ligands to their cognate receptors plays a crucial
role in several fundamental biological processes, including apoptosis, cellular differentiation,
and inflammation [11]. TNF-like weak inducer of apoptosis (TWEAK, encoded by the
*TNFSF12* gene) is a member of the TNF ligand superfamily originally described in 1997 [12].
A specific receptor for TWEAK, fibroblast growth factor-inducible 14 (Fn14), is encoded by
the *TNFRSF12A* gene and was conclusively identified in 2001 [13]. TWEAK is a multifunc-
tional proinflammatory cytokine that regulates cell proliferation, migration, survival, differen-
tiation, and death via its highly inducible receptor, Fn14 [14–17]. Upregulation of Fn14 after
various acute tissue insults promoted tissue responses involved in regeneration and repair, for
example, in mouse skeletal muscle precursor cells [18] and in the human liver [19]. Moreover,
Nazeri et al. suggested that the continuous or excessive activation of TWEAK/Fn14 signaling
contributes to promoting damage and chronic inflammation in diseased tissues with multiple
sclerosis and experimental autoimmune encephalomyelitis [20]. Recent evidence has suggested
that TWEAK/Fn14 also play important roles in acute kidney injury [14] and in experimental
and human chronic kidney disease [21].

In the kidney, Fn14 expression was observed in tubular cells in cultured murine renal tubu-
lar cells and in mice, and it was upregulated by proinflammatory cytokines [22]. Gao et al.

reported Fn14 expression on the surface of human mesangial cells, podocytes, and renal tubular cells by flow cytometric analysis [23]. TWEAK, in addition, has proinflammatory effects on glomerular mesangial cells in mice [24], inducing the expression of inflammatory cytokines, such as monocyte chemoattractant protein-1 (MCP1, CCL2), interleukin (IL)-6, RANTES, and CXCL16, and downregulating the expression of Klotho [25–28].

The role of the TWEAK/Fn14 system in the pathogenesis of IgAN, however, is unclear. Few reports have described an association between TWEAK/Fn14 and IgAN. Sasaki et al. reported a relationship between the levels of urinary TWEAK and clinicopathological findings and suggested that the TWEAK/Fn14 system might affect crescent formation and proteinuria in adult patients with IgAN [29]. However, no study has reported the role of TWEAK/Fn14 in pediatric IgAN.

Recently, a high IgA strain (HIGA) of ddY mice obtained by selective mating of ddY mice with high serum IgA was established as an inbred murine model of IgAN [30, 31]. Moderate to severe mesangial IgA deposition and progressive mesangial matrix accumulation were observed in HIGA mice [31]. Subsequent studies used HIGA mice to elucidate the pathogenesis of IgAN [32–34].

In this study, we examined whether there was an association between Fn14 expression in glomeruli and disease activity in clinical and histological findings of pediatric IgAN. We investigated the clinical features, histological findings, and patterns of gene expression in high serum IgA (HIGA) mice, a murine model of IgAN, using a gene expression array with a next-generation sequencer. Through these analyses, we clarified the roles of Fn14 in the pathogenesis of pediatric IgAN and determined whether TWEAK/Fn14 can be used as a marker to predict disease progression.

## Materials and methods

### Patient samples

Formalin-fixed paraffin-embedded tissue samples from 14 patients with pediatric IgAN were obtained by renal biopsy at our institution between 2008 and 2016. The study protocol was in accordance with the standards of the ethics committee of the institution. In accordance with the Declaration of Helsinki, we explained to the guardians of the subjects enrolled in our study and obtained their written informed consent. All research was approved by the institutional review board at Ehime University (approval no. 30-16-R2-K9). The clinical characteristics of the patients are summarized in Table 1. No steroids or immunosuppressants were used by patients prior to the time of renal biopsy. The patients were categorized based on the "Guidelines for the Treatment of Childhood IgAN" proposed by the Japanese Society for Pediatric Nephrology [35], as follows: "severe" type was defined as disease with heavy proteinuria (early morning urinary protein/creatinine ratio of 1.0 or higher) or diffuse lesions, more than 80% of glomeruli, with mesangial proliferation, crescent formation, adhesion, or sclerosis. "Mild" type was defined as disease without the criteria for the severe type. Ten patients (cases 1–10) were categorized as having mild disease, and four patients (cases 11–14) were categorized as having severe disease. Regarding the Oxford classification of IgA nephropathy [3, 36], as shown in Table 1, all of the mild type patients presented M0 and all of the severe type patients presented M1. None of the patients had tubulointerstitial lesions >25%, T1, or T2. Case 14 was diagnosed with nephrotic-nephritic IgAN. No patients progressed to ESRD; however, persistent proteinuria continued in two of four patients with severe disease.

The expression of Fn14 was examined by immunohistochemistry and immunofluorescence, and the association between Fn14 expression and clinical features was evaluated. Fn14 expression in glomeruli was evaluated semi-quantitatively by a single investigator blindly, as

**Table 1. The clinical characteristics of pediatric patients with IgAN in this study.**

| No. | Sex Age (yr) | Type [a]/ Oxford [b] MESTC | Initial symptoms and Upro/Cr at biopsy | Period[c] and Treatment prior to biopsy | Additional therapy | Prognosis and follow-up period [d] |
|---|---|---|---|---|---|---|
| 1 | M | Mild/ | SS | 9 mo | MPT | CR |
|  | 12 | M0E0 | 0.14 | ACE-I | Tonsillectomy | 5 yr |
|  |  | S0T0C0 |  |  |  |  |
| 2 | M | Mild/ | MH | 14 mo | MPT | CR |
|  | 9 | M0E0 | 0.27 | ACE-I | Tonsillectomy | 4.5 yr |
|  |  | S0T0C1 |  |  |  |  |
| 3 | F | Mild/ | MH | 5 mo | ACE-I | CR |
|  | 9 | M0E0 | 0.25 | None | MPT | 4.5 yr |
|  |  | S0T0C0 |  |  | Tonsillectomy |  |
| 4 | M | Mild/ | MH | 8 mo | ACE-I | CR |
|  | 13 | M0E0 | 0.42 | anti-platelets |  | 3.6 yr |
|  |  | S0T0C0 |  |  |  |  |
| 5 | F | Mild/ | MH | 6 mo | ACE-I | CR |
|  | 7 | M0E0 | 0.14 | None | MPT | 5.2 yr |
|  |  | S0T0C0 |  |  | Tonsillectomy |  |
| 6 | M | Mild/ | Chance | 5 mo | None | CR |
|  | 6 | M0E0 | 0.34 | ACE-I |  | 1.6 yr |
|  |  | S0T0C0 |  |  |  |  |
| 7 | F | Mild/ | SS | 8 mo | ACE-I | Hematuria |
|  | 12 | M0E0 | 0.45 | anti-platelets | MPT | (mild) |
|  |  | S0T0C0 |  |  | Tonsillectomy | 4.7 yr |
| 8 | F | Mild/ | SS | 10 mo | ACE-I | CR |
|  | 11 | M0E0 | 0.31 | None |  | 3 yr |
|  |  | S0T0C0 |  |  |  |  |
| 9 | M | Mild/ | Chance | 6 mo | ACE-I | CR |
|  | 8 | M0E0 | 0.23 | None | prednisolone | 4.6 yr |
|  |  | S0T0C0 |  |  |  |  |
| 10 | F | Mild/ | SS | 12 mo | ACE-I | CR |
|  | 13 | M0E0 | 0.32 | anti-platelets |  | 3.2 yr |
|  |  | S0T0C0 |  |  |  |  |
| 11 | M | Severe/ | SS | 34 mo | Combination [e] | Hematuria |
|  | 6 | M1E1 | 1.44 | ACE-I | MPT | (mild) |
|  |  | S0T0C0 |  |  | Tonsillectomy | 6.2 yr |
| 12 | F | Severe/ | SS | 2 mo | Combination [e] | Persistent |
|  | 9 | M1E0 | 2.28 | ACE-I | MPT | proteinuria |
|  |  | S0T0C1 |  |  | Tonsillectomy | (mild) 7.4 yr |
| 13 | M | Severe/ | SS | 4 mo | Combination [e] | CR |
|  | 11 | M1E1 | 0.67 | None | ACE-I | 6.3 yr |
|  |  | S0T0C1 |  |  | MPT |  |
|  |  |  |  |  | Tonsillectomy |  |
| 14 | M | Severe/ | SS[f] | 42 mo | Combination [e] | Persistent |
|  | 13 | M1E1 | with MH, |  | ACE-I, Cy-A | proteinuria |
|  |  | S1T0C2 | Nephrotic | None | MPT | (moderate) |
|  |  |  | 7.84 |  | Tonsillectomy | 5.5 yr |

[a]Type based on the Guidelines for the Treatment of Childhood IgAN.

[b]Oxford classification of IgA nephropathy.

[c]Period to biopsy since the first urinalysis abnormality was pointed out.

[d]Follow-up period; from renal biopsy performed.

[e]Combination therapy containing prednisolone, mizoribine, anti-platelets and anti-coagulant drugs.

[f]Case 14 was first identified as a mild urinary disorder in the school screening, then detected with heavy proteinuria in the school screening 38 months later, and diagnosed with macrohematuria and nephrotic syndrome at the first visit to our hospital.

yr, years old, at renal biopsy; mo, months; Upro/Cr, ratio of urinary protein/urinary creatinine; SS, school urinary screening; MH, macrohematuria; Chance, chance urinalysis; CR, complete remission; ACE-I, angiotensin converting enzyme-inhibitor; MPT, methyl-prednisolone pulse therapy; Cy-A, cyclosporin-A.

follows: 0, no staining of tufts; 1, staining within 25% of tufts; 2, staining in 25%–50% of tufts; 3, staining in 50%–75% of tufts; 4, staining in more than 75% of tufts.

## Animal models

HIGA mice, established by the selective mating of ddY mice as described previously, are animal models of the spontaneous development of mesangio-proliferative glomerulonephritis with high serum IgA, mimicking human IgAN [30, 31]. Female HIGA mice (Japan SLC, Inc., Shizuoka, Japan) were used for the analysis. Because hemi-nephrectomy enhances the development of nephropathy, as previously reported in another mouse model [34], we performed left hemi-nephrectomy under inspiratory anesthesia to accelerate glomerular injury at 6 weeks of age, as follows. Isoflurane inhalation anesthetic was used for introduction at a concentration of 4% isoflurane in a dedicated gauge, while maintaining the concentration at 0.5% with a mask following loss of consciousness. The mice were placed in the prone position and disinfected with alcohol prior to the surgical procedure. The incision was made in the midline, slightly caudal to the erector spinae muscle. The left latissimus dorsi muscle was opened from the side of the erector spinae muscle, and the left kidney was peeled off to avoid damage to the adrenal gland. The left kidney was dissected and excised by ligation with a 3–0 silk thread on the median side. The latissimus dorsi muscles were sutured together and the skin was sutured together. After nephrectomy, the mice were bred with a gauge different from that of the other mice until they were awake and moved around.

After hemi-nephrectomy, the mice were sacrificed at 9, 11, 14, 26, or 40 weeks of age by cervical dislocation, while under anesthesia, by a skilled experimenter. Kidneys were collected for further analysis. Female, age-matched BALB/c mice were used as controls in the analysis, and the blinding procedure was not used for this experiment. Three HIGA mice and two control mice were analyzed each week because of the minimum number of samples. All mice alive at the analysis time point were used for the analysis, and 18 HIGA mice and 18 BALB/c control mice were used in the experiment.

We measured the body weight of each mouse twice a week during the experiment to monitor their health state and to monitor whether there was a sudden weight gain or loss of more than 50% in 1 week. We carefully observed the activity of the mice in the gauge and the amount of food reduced every day.

All animal experiments were performed in accordance with the guidelines of the Institutional Animal Care and Use Committee of Ehime Graduate School of Medicine, under the approval of the Committee (approval no. 05-TE21-1).

## Measurement of urinary protein excretion and serum markers in mice

Spot urine samples were obtained from all mice weekly. The mice were moved one by one from the breeding gauge to the urine collection container, and urine was collected with a dropper during spontaneous urination. In the absence of spontaneous urination, the lower abdomen of the mice was massaged in a protective manner to encourage urination. The concentrations of urinary protein and creatinine were measured at the Nagahama Life Science Laboratory (Oriental Yeast Co., Ltd, Tokyo, Japan) with urine samples suitable for measurement. The ratio of urinary protein to creatinine was calculated for four HIGA mice and three control mice. Serum was also collected from mice at the time of sacrifice, and serum levels of creatinine and albumin were measured using a biochemical autoanalyzer (Dri-chem 3500V; Fujifilm, Tokyo, Japan).

## Pathological analysis

Renal tissues were fixed in neutral 10% buffered formalin solution and embedded in paraffin for examination. Three-micron-thick sections were prepared and stained with hematoxylin-

eosin or periodic acid-Schiff (PAS). The stained renal tissues were examined by an experienced pediatric nephrologist (Y.T.) under a light microscope, and all obtained findings were reviewed by other researchers (M.E-I., T.I., and M.E.). The cellular crescent formation score was calculated as the percentage of glomeruli with cellular crescent formation among all glomeruli. We also evaluated cell counts and areas in 20 randomly selected glomeruli.

## Immunohistochemistry

Immunohistochemical analysis of IgA and Fn14 was performed as follows: Three-micron-thick paraffin-embedded kidney sections were dewaxed, rehydrated, and treated with 3% $H_2O_2$ for 10 min to inhibit endogenous peroxidase activity. The sections were pretreated by boiling in 10 mM citrate buffer (pH 6.0) for 6 min to enhance their immunoreactivity. Next, the sections were treated with Dako Real Antibody Diluent (Dako, Santa Clara, CA, USA) for 40 min to prevent nonspecific antibody binding. Sections were incubated overnight at 4°C with primary antibodies, followed by incubation with the appropriate secondary antibodies. Staining was then visualized with DAB as a substrate, and hematoxylin was used for counterstaining.

The antibodies used for the study were as follows: polyclonal goat anti-mouse IgA antibodies conjugated with horseradish peroxidase (HRP; Bethyl Laboratories, Montgomery, USA; 1:500 dilution) and monoclonal rabbit anti-TWEAKR (Fn14) antibodies (Abcam, Cambridge, UK; 1:100 dilution) as primary antibodies and DAKO Envision+System-HRP-labeled polyclonal anti-rabbit IgG as a secondary antibody for Fn14 staining.

The stained renal tissues were examined by an experienced pediatric nephrologist (Y.T.) under a light microscope, and all findings obtained were reviewed by other researchers (M. E-I., T.I., and M.E.). Images of positively stained glomeruli were acquired, and the proportion of the Fn14-stained area in each glomerulus was calculated using the ImageJ software (https://imagej.nih.gov/ij/). Statistical analyses were performed by the Mann-Whitney U test with JMP software (SAS Institute Japan Ltd., Tokyo, Japan).

## Tissue sampling by laser-captured microdissection

Kidney tissues were immediately frozen in OCT compound (Sakura Finetek Japan, Tokyo, Japan), and 12-μm-thick cryostat sections were prepared and mounted onto foil-coated glass slides (2.0 μm PEN-Membrane slides; Leica Microsystems, Wetzlar, Germany; cat. no. 11505189). The sections were immediately fixed with 70% ethanol for 1 min and washed with RNase-free water for 5–10 s. The sections were then stained rapidly with 0.05% toluidine blue stain for 30–60 s, washed twice with RNase-free water, and air dried with a fan for approximately 10 min.

Laser-captured microdissection was performed using the application solution laser capture microdissection system (Leica Microsystems). After tracing the targeted glomeruli, we dissected the glomeruli together with the surrounding thin membrane using a laser microbeam and placed the samples into a microcentrifuge cap. We dissected more than 500 glomeruli (1,000 glomeruli, where possible) from each mouse and pooled them for RNA extraction.

## RNA extraction

Total RNA was extracted from the glomeruli collected by laser-captured microdissection using an Agencourt RNAdvance Tissue Kit (Beckman Coulter, USA) according to the manufacturer's instructions. The integrity of the total RNA was assessed using a BioAnalyzer RNA6000 pico kit (Agilent Technologies, Palo Alto, CA, USA). RNA integrity number (RIN) of extracted RNA was between 2 and 4 in all samples, indicating fragmentation of RNA due to microdissected tissue samples as the source of RNA.

## Gene expression array based on whole transcriptome (RNA-Seq) analysis

First-strand cDNA was synthesized and purified from total RNA using the SMARTer Stranded Total RNA-Seq Kit-Pico Input Mammalian (Takara Bio Inc., USA) according to the manufacturer's instructions. The quality of the library was assessed using a Bioanalyzer High-Sensitivity DNA kit. RNA sequencing was performed using the MiSeq Reagent Kit V3 150 cycle kit (Illumina, USA). The RNA-Seq data obtained were analyzed using Tophat, featureCounts, and R software. Raw read counts were normalized and scaled using trimmed mean of M-values (TMM) normalization, and differential expression analysis was performed by exact test using the edgeR package. Differentially expressed genes were selected based on a false discovery rate (FDR) cutoff of 5% and a fold change >2 or <0.5. Gene expression data have been deposited in the DDBJ database (accession number DRA012541).

# Results

## Fn14 is expressed in the early phase of pediatric IgAN

Renal tissues obtained from 14 patients (eight boys and six girls) with pediatric IgAN were evaluated by immunohistochemistry analysis for Fn14 expression, which was evaluated and categorized semiquantitatively into five grades. The results for each patient are shown in Fig 1A. Glomeruli with marked Fn14-positive staining (grades 3–4) were observed more frequently in mild IgAN cases, M0E0S0T0C0 or M0E0S0T0C1 on Oxford classification, than in severe cases with M1. In addition, staining of Fn14 in glomeruli collected from severe cases was relatively weak compared with that in mild cases, and even glomeruli with crescent formation did not show Fn14-positive staining. Generally, in mild and severe cases, Fn14 expression was not observed in glomeruli with obvious mesangial proliferation and crescent formation, but was observed in the glomerular tuft with an intact appearance. This suggests that Fn14 plays a role in the early phase of IgAN pathogenesis (Fig 1B).

## Development of nephropathy in HIGA mice after hemi-nephrectomy

Left hemi-nephrectomy of HIGA mice was conducted at 6 weeks of age to enhance glomerular injury in the remaining kidney. After hemi-nephrectomy, the mice were sacrificed at 9, 11, 14, 26, or 40 weeks of age, and residual kidneys were collected for further analysis (Fig 2A). Urinary protein excretion from HIGA mice with hemi-nephrectomy was evaluated as the ratio of protein/creatinine (U-pro/Cr) after hemi-nephrectomy from 6 to 40 weeks of age and compared with U-pro/Cr of control mice (n = 3). Fig 2B shows the ratio of U-pro/Cr in HIGA mice to that in control mice, and urinary protein excretion in HIGA mice increased from 14 weeks to 40 weeks, peaking at 26 weeks of age. The U-pro/Cr ratio of control mice was stable throughout the experiment. Pathological changes were evaluated by light microscopy examination of the kidneys collected after sacrifice. Light microscopy images of PAS-stained kidney tissues in HIGA mice at 6, 9, 11, 14, 26, and 40 weeks of age are shown in Fig 2C. Mesangial proliferative lesions in glomeruli were not obvious in HIGA mice up to 11 weeks of age. After 14 weeks, mesangial matrix proliferation was more evident than mesangial cell proliferation, particularly at 40 weeks, and apparent proliferation of the mesangial matrix was observed.

## Fn14 expression in HIGA mice is present during the early phase of nephropathy

Deposition of IgA and the expression of Fn14 in the glomeruli of HIGA mice were analyzed beginning at 6 weeks of age, before the proliferation of mesangial matrix became evident. IgA-

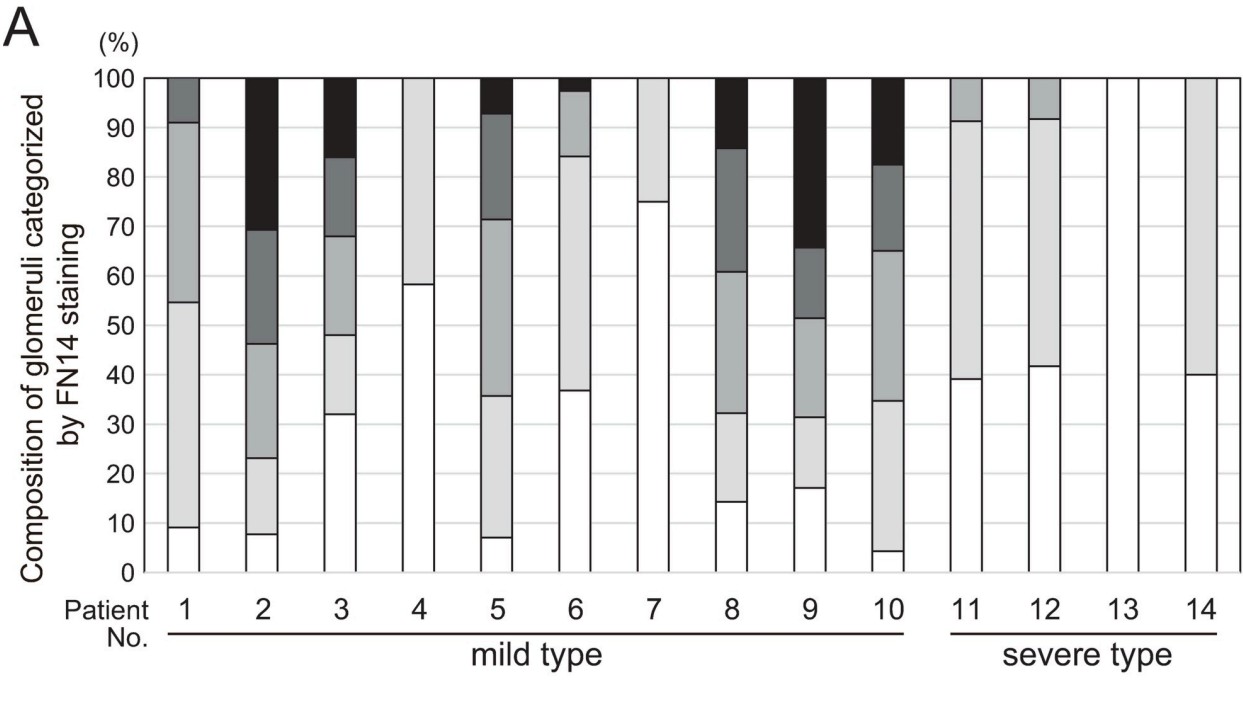

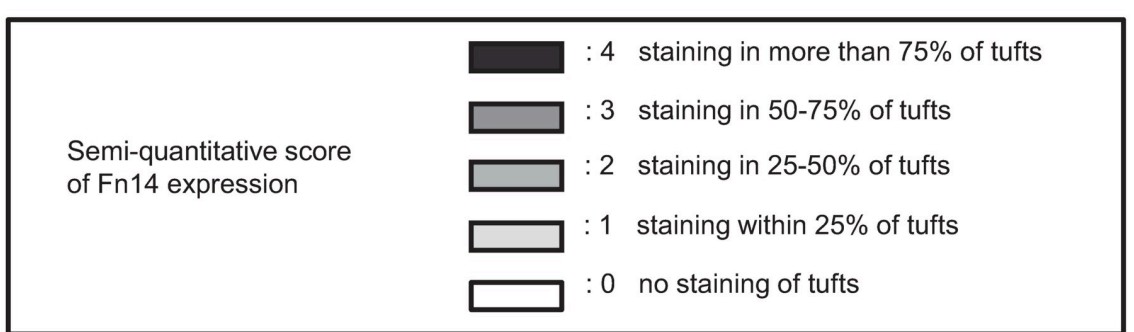

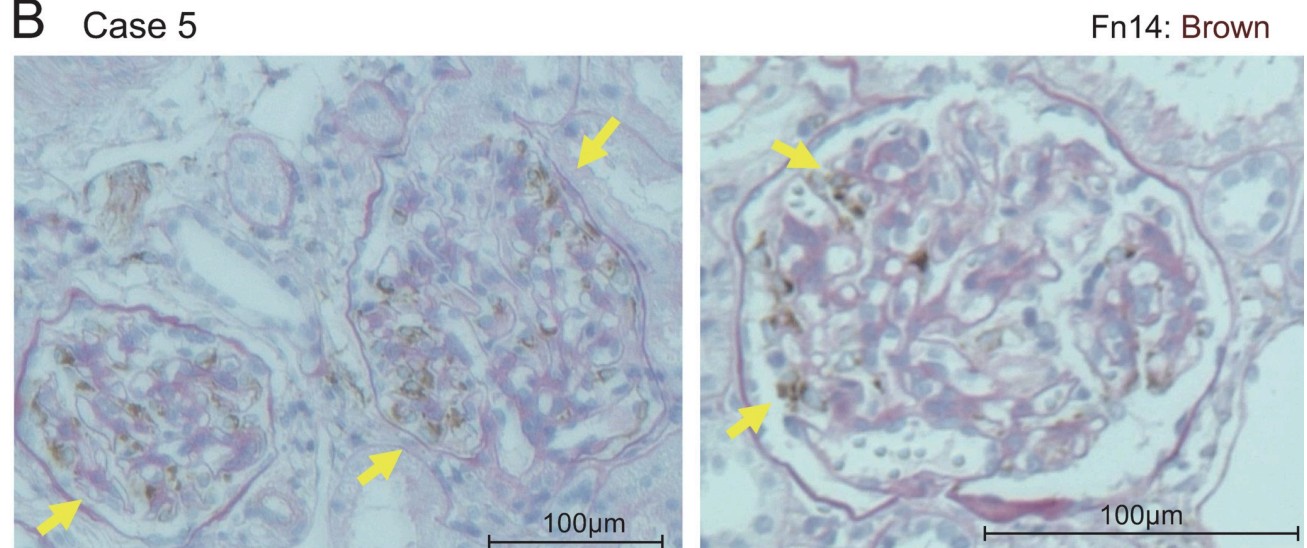

**Fig 1. Fn14 expression in glomeruli from biopsied specimens obtained from patients with IgAN.** (A) Fn14 expression was evaluated semiquantitatively in each patient. Ten patients with mild disease (cases 1–10) and four patients with severe disease (cases 11–14) were analyzed. Semiquantitative scoring of Fn14 positivity was categorized as follows: 0, no staining of tufts; 1, staining within 25% of tufts; 2, staining in 25–50% of tufts; 3, staining in 50–75% of tufts; and 4, staining in more than 75% of tufts. (B) Fn14 staining was frequently detected in glomeruli with an intact appearance. Double PAS staining and Fn14 immunostaining in a case with mild IgAN (case 5 in Table 1). Two representative glomeruli are shown. Fn14 deposits existed segmentally in one glomerulus and were detected in glomeruli showing an intact appearance without any mesangial proliferation in a patient with mild disease. This patient had mild proteinuria and repeated macrohematuria during upper respiratory infection and enterocolitis.

positive glomeruli were not observed in HIGA mice at 6 weeks of age; however, IgA-positive glomeruli were detectable in a few glomeruli collected from HIGA mice at 9 weeks of age. IgA-positive glomeruli were stably detectable thereafter, beginning at 11 weeks, and were more frequently observed (with intense staining) after 14 weeks of age in HIGA mice (Fig 3A). Glomeruli in control mice rarely showed IgA positivity throughout the study until 40 weeks of age. In HIGA mice and control mice, Fn14 was detected in renal tubular cells, consistent with previous reports [11]. The expression of Fn14 in glomeruli was not detectable from 6 to 9 weeks in HIGA mice and control mice, but was present at 11 weeks of age, and more intensely positive glomeruli were frequently observed in 14-week-old HIGA mice (Fig 3B). The intensity of Fn14 staining in the glomeruli was measured on dissected kidney specimens obtained from 6-, 9-, 11-, 14-, 26-, and 40-week-old HIGA mice by acquiring and analyzing the stained images using ImageJ software, and the results were compared. As shown in Fig 3C, the intensity of Fn14 staining increased from 11 weeks, peaked at 14 weeks with a significant difference, and then diminished at 26 weeks. Fn14 staining was also observed in glomerular tufts of glomeruli without crescent formation.

## Gene expression array by RNA-Seq analysis of laser-dissected glomeruli from HIGA mice

Next, we performed gene expression array analysis based on the transcriptome profiles obtained from RNA-Seq analysis. Laser-microdissected glomeruli were collected from three HIGA mice and three control mice at 6, 11, and 26 weeks of age and analyzed. The expression of *Fn14* in glomeruli of HIGA mice was high at 6 weeks (indicating the pre-onset phase), and at 11 weeks (early phase) was slightly increased compared with that at 6 weeks. The expression of *Fn14* at 26 weeks, however, decreased to the control levels. *Tweak* expression levels in HIGA mice were similar to those in the control mice (Fig 4A). Additionally, we analyzed 87 target genes downstream of *Fn14*. The changes in several downstream targets of *Fn14* are summarized in Fig 4B. The expression patterns of target genes were altered at each stage of the disease development. Genes downregulated more than 2-fold at 11 weeks compared with that at 6 weeks (Fig 4C) included *Mapk 15*, *Mapk 10*, *Mapk 13*, *Traf3ip3*, and *Traf1*. Genes upregulated more than 2-fold at 11 weeks compared with levels at 6 weeks (Fig 4D) included *Nfkbib*, *Nfkbid*, *Rac2*, *Rac3*, *Wnt5b*, and *Wnt7b*.

Across the whole transcriptome analysis, 77 genes showed more than a 4-fold increase in HIGA mice at 11 weeks of age compared with that at 6 weeks of age, and there were no changes in control mice. Additionally, 109 genes were downregulated by more than 4-fold in glomeruli from HIGA mice obtained at 11 weeks compared with those at 6 weeks. The differentially expressed genes at these time points are summarized in Fig 5. Characteristic upregulation of some gene sets was observed at each time point, and these genes were involved in several biological pathways, as shown in Fig 5A and 5B. Many genes were upregulated at 11 weeks of age in pathways including matrix proteins, TNF signaling, signal transduction, and immune network for IgA production. Furthermore, some genes in the complement pathway were upregulated after 26 weeks.

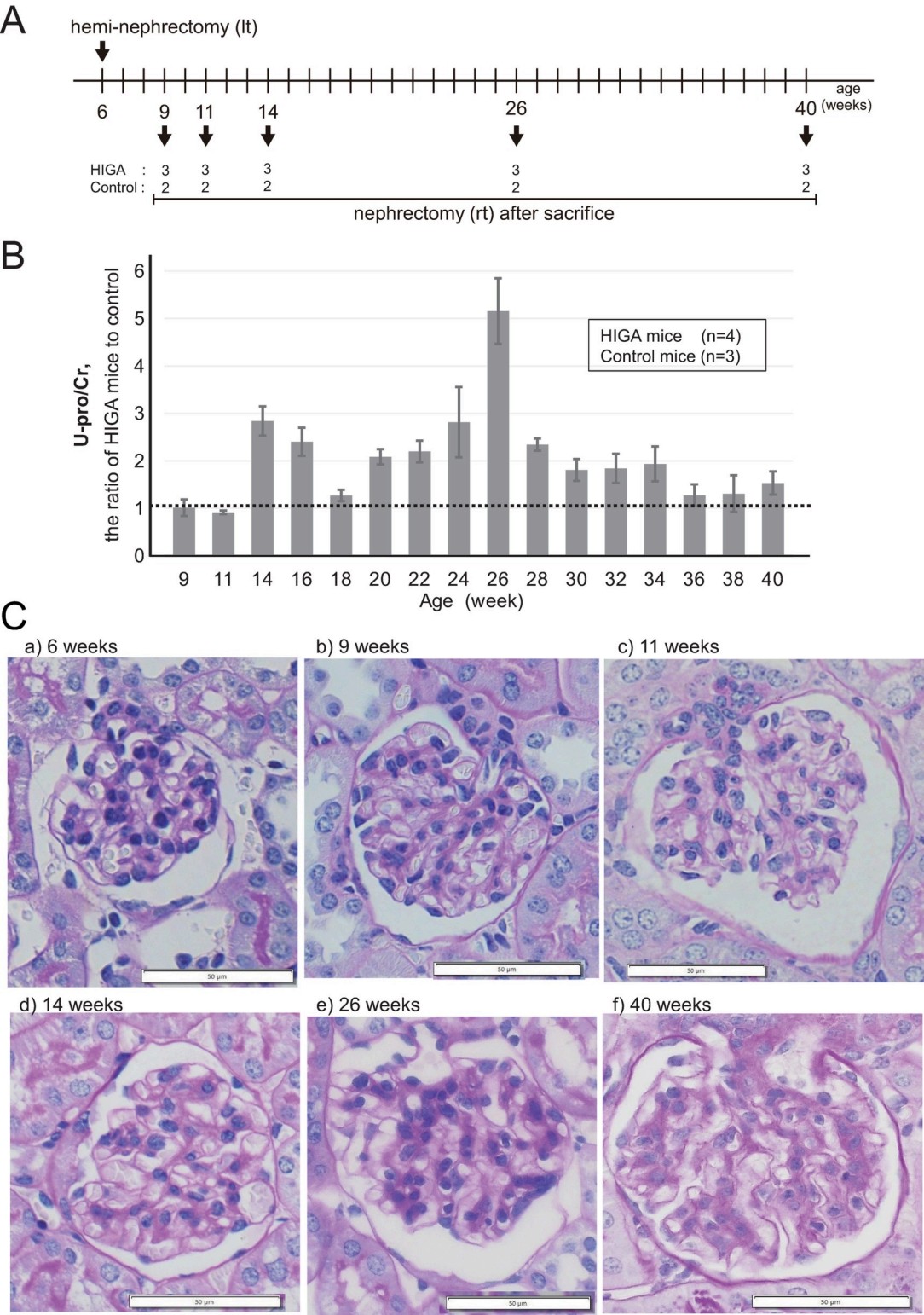

**Fig 2. Development of IgA nephropathy in HIGA mice with hemi-nephrectomy.** (A) Time course of IgA nephropathy HIGA mice. Left hemi-nephrectomy under inspiratory anesthesia was performed to accelerate glomerular injury at 6 weeks of age. After hemi-nephrectomy, the mice were sacrificed at 9, 11, 14, 26, or 40 weeks of age, and the right residual kidney was collected for further analysis. (B) Urinary protein excretion of hemi-nephrectomized HIGA mice. The mean U-pro/Cr ratios in HIGA (n = 4) and control mice (n = 3) are shown. U-pro/Cr tended to be increased in HIGA mice from 14 weeks of age, with a peak at

26 weeks of age. (C) Pathological observations in HIGA mice after hemi-nephrectomy. Light microscopy findings (PAS staining) in HIGA mice are shown. Mesangial proliferative lesions were not obvious at 6 (a), 9 (b), and 11 weeks (c). After 14 weeks, the proliferation of mesangial matrix was more evident than the mesangial cell proliferation, particularly at 40 weeks (f), apparent proliferation of the mesangial matrix was observed. 200× magnification.

## Discussion

In Japan, all children between 6 and 18 years of age are screened annually for the presence of asymptomatic hematuria/proteinuria, facilitating the early diagnosis of IgAN within 1 year of the actual onset of the disease [7–9]. In this study, we aimed to identify the factors involved in the early phase of IgAN pathogenesis.

The TWEAK/Fn14 pathway has various proinflammatory roles, including the regulation of cell proliferation, migration, survival, differentiation, and death [14–17]. Moreover, the TWEAK/Fn14 system is involved in the activation of glomerular mesangial cells and in podocyte-induced proliferation and inflammatory cytokine production [23, 24]. In adult IgAN, TWEAK/Fn14 was reported to be associated with crescent formation [29]. Thus, we evaluated the role of the TWEAK/Fn14 system in the pathogenesis of pediatric IgAN.

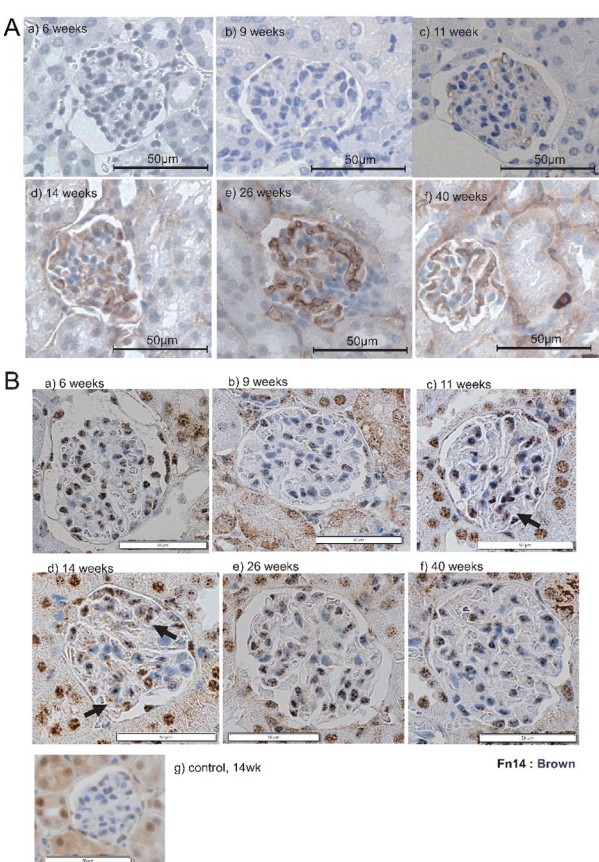

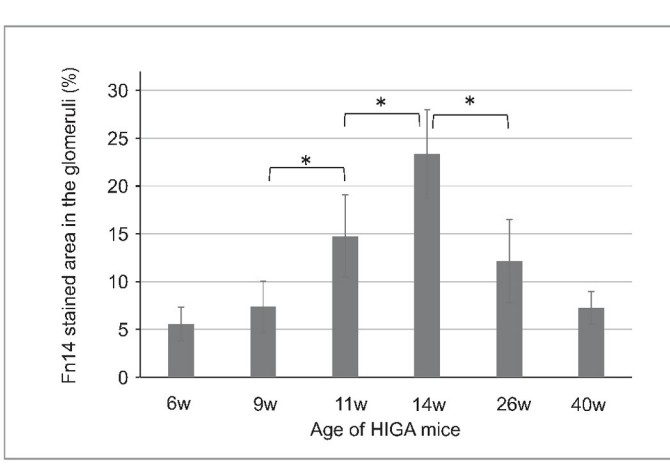

**Fig 3. Pathological observations in HIGA mice after hemi-nephrectomy.** (A) Immunohistochemical staining with IgA. IgA deposition was not observed at 6 or 9 weeks (a, b). Slightly IgA-positive glomeruli were detected at 11 weeks (c). IgA-positive glomeruli were frequently detectable after 14 weeks (d–f). 200× magnification. (B) Immunohistochemical staining for Fn14. Positive Fn14 staining in glomeruli was observable at 6 and 9 weeks (a, b). Fn14 deposition became more obvious at 14 weeks (d). Fn14 deposition was not evident at 26 or 40 weeks (e, f). Negative Fn14 staining in glomeruli of a control mouse (g). Arrows indicate positive Fn14 staining in glomeruli. (C) The proportion of the Fn14 stained area in the glomeruli at different time points. The average values of 10 glomeruli are shown at 6, 9, 11, 14, 26, and 40-week-old HIGA mice. *: p<0.001(Mann-Whitney U test).

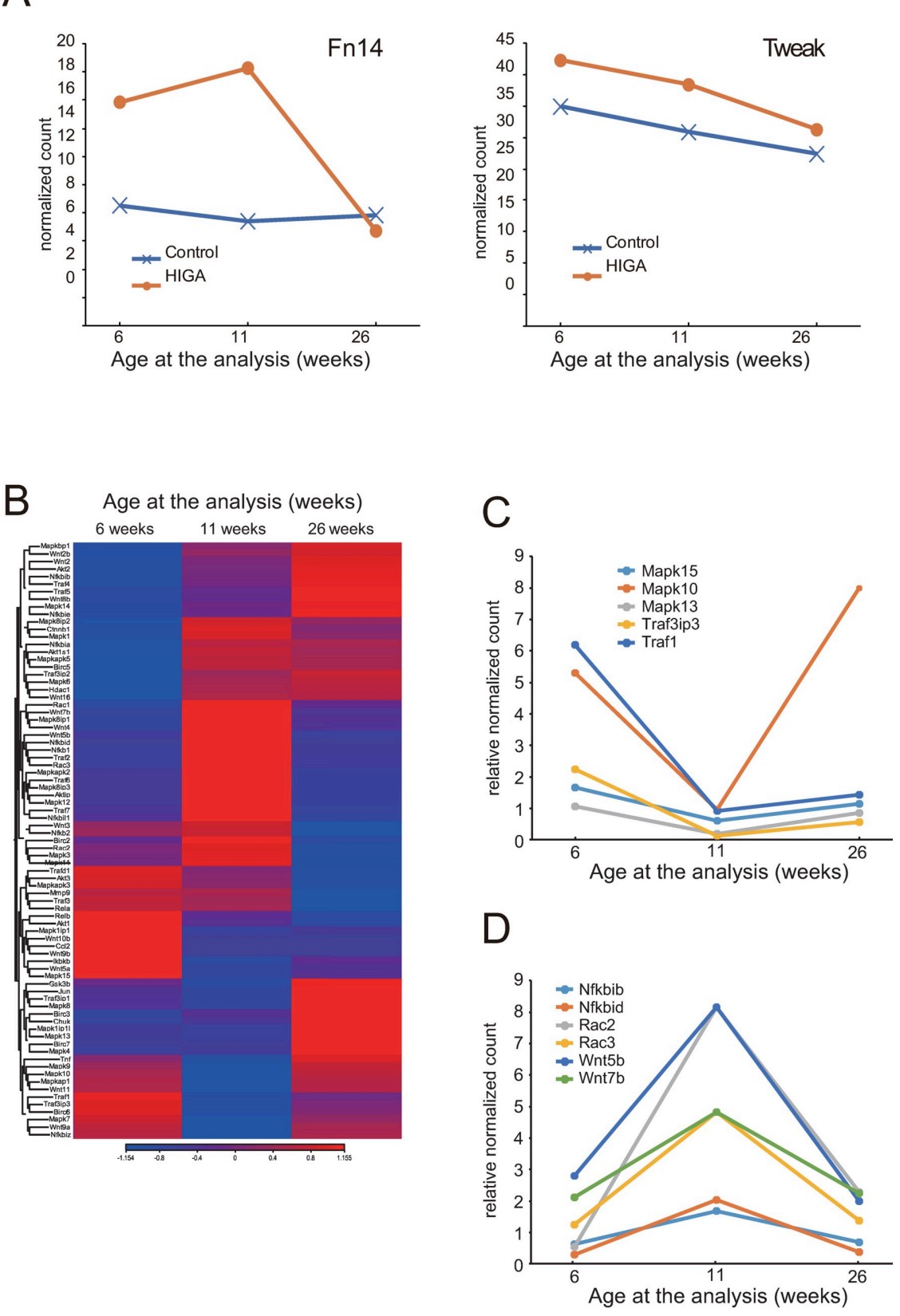

**Fig 4. Expression patterns of *Fn14, Tweak* and their downstream targets in the glomeruli of HIGA mice.** (A) Expression levels of *Fn14* and *Tweak* in glomeruli obtained from control and HIGA mice at each time point. Normalized counts of each gene are used to indicate the expression level. (B) Comparison of the expressions of genes reported to be regulated by the Tweak/Fn14 pathway. (C, D) Representative pattern of gene expression is shown.

Pathological analysis of biopsied kidney tissues from 14 patients, including 10 mild cases and four severe cases, showed that intense Fn14 deposition in glomeruli was observed in 80% of mild cases, but was not observed in severe cases. Interestingly, Fn14-positive glomeruli were found even in mild cases, with a score of almost 0 based on the Oxford classification. In addition, Fn14 expression was not present with obvious mesangial proliferation or in the crescent region of glomeruli, but was detected strongly in intact glomerular tufts. These findings differed from those of a previous report of adult cases of IgAN [29]. No patients were administered steroids or immunosuppressants prior to the time of the renal biopsy in our study; therefore, they were considered unlikely to involve inflammatory cytokines. We speculate that Fn14 acts on glomeruli during a relatively early stage of IgAN in childhood. However, we analyzed the expression of Fn14 in only 14 pediatric cases, comparing four severe cases with 10 mild cases. This number of patients was too small to draw meaningful conclusions, and we could not perform transcriptome analysis of patient biopsy samples due to a shortage of available samples for the analysis.

HIGA mice, a mouse model of IgAN, have high serum IgA levels and undergo spontaneous development of mesangio-proliferative glomerulonephritis [30]. Because hemi-nephrectomy enhances the development of nephropathy [37], all mice underwent left hemi-nephrectomy at 6 weeks of age. To determine the time course of disease progression in HIGA mice, particularly during the early stages of disease, we analyzed urinary protein excretion and immunopathology. As shown in Fig 2B, urinary protein excretion in HIGA mice increased from 14 to 40 weeks, peaking at 26 weeks of age. Intense IgA deposition in glomeruli was observed at 14 weeks of age; thereafter, the proliferation of mesangial substrates became evident, and cell density in the glomerular capillaries decreased at 26–40 weeks of age, indicating an advanced stage of nephropathy. We speculate that the pathological process of IgAN may begin at approximately 14 weeks of age in HIGA mice. However, any increase in proteinuria might be related to the pathological progression of IgAN, as well as hyperfiltration with the hemi-nephrectomy procedure. In this study, we did not compare hemi-nephrectomy HIGA mice with those without hemi-nephrectomy.

Fn14 deposition was detectable at 11 weeks of age, when urinary protein excretion had not yet increased, and more intense Fn14 staining was observed at 14 weeks of age. Fn14 staining became weaker at 26 weeks with increased urinary protein excretion and undetectable at later points, such as 40 weeks. These observations obtained from the mouse model also indicate that Fn14 may act at an early stage of IgAN pathogenesis.

Fn14 was reported to be associated with crescent formation in adult IgAN cases [29]. In our study, crescent formation was only observed in one mouse at 14 weeks of age; therefore, we could not evaluate the association between Fn14 and crescent lesions.

To study the expression of downstream targets in the TWEAK/Fn14 pathway and to identify other factors involved in the pathogenesis of IgAN, gene expression array analysis based on RNA-Seq was performed using laser-captured microdissected glomeruli obtained from HIGA mice at several time points (6, 11, and 26 weeks of age). The expression of *Fn14* was higher than that of controls at 6 weeks of age and was slightly increased at 11 weeks of age compared with that at 6 weeks of age. This decrease was observed at 26 weeks of age. *Fn14* gene expression was upregulated in the early phase of IgAN, similar to Fn14 deposition in glomeruli. This indicated that Fn14 is associated with disease pathogenesis during the early phase of IgAN, when evident mesangial proliferation and proteinuria are not yet initiated.

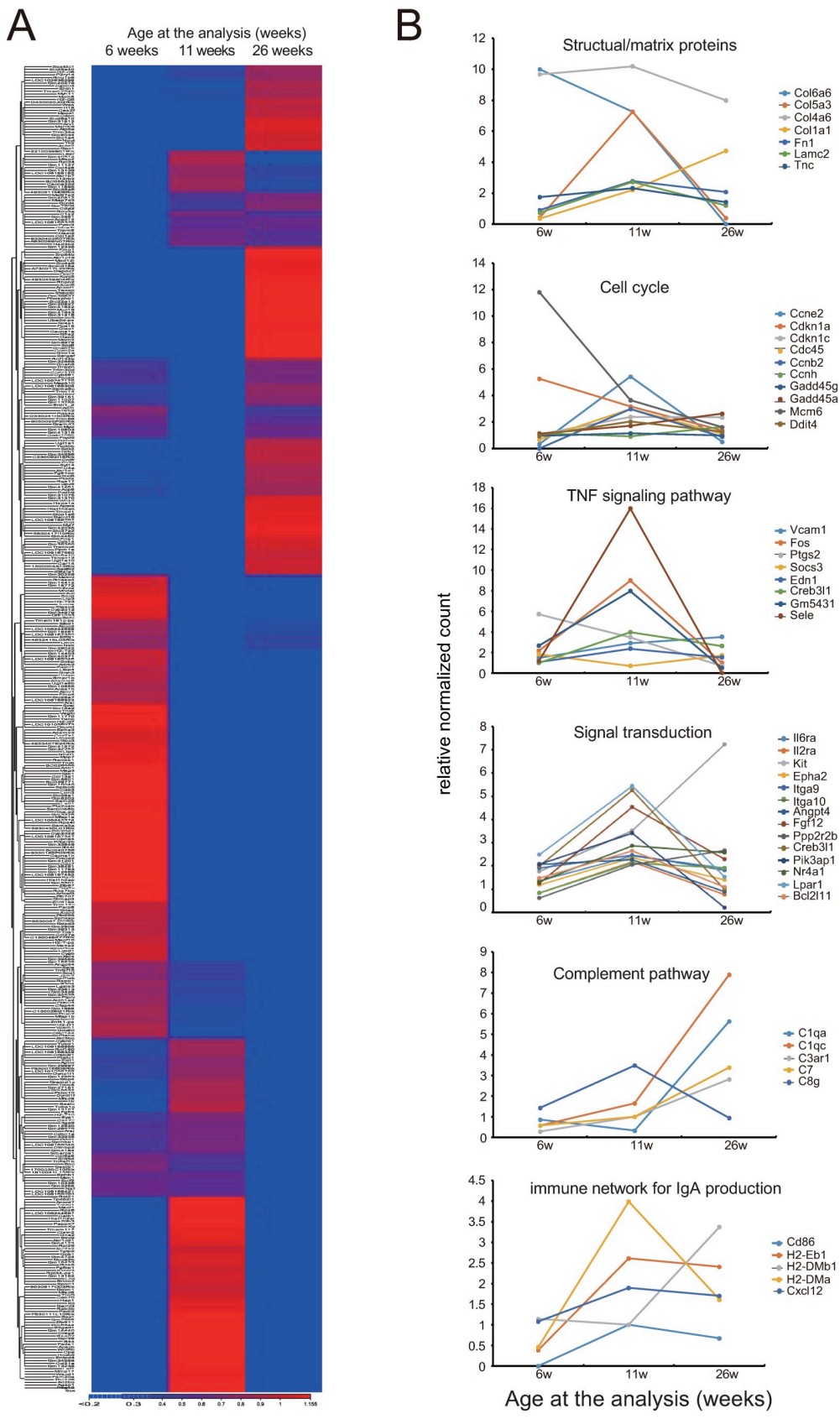

**Fig 5. Expression pattern of differentially expressed genes in glomeruli obtained from HIGA mice.** (A) Differentially expressed genes in the glomeruli of HIGA mice at each time point. (B) Several biological pathways were identified from these differentially expressed gene sets. Representative pathways and expression patterns of the involved genes are shown. Ratio of the normalized count of HIGA mice to control mice was used as a relative normalized count.

Next, we investigated the downstream targets of Fn14 in the pathogenesis of IgAN. We analyzed changes in the expression of downstream targets of TWEAK/Fn14. Among the downstream targets, *Mapk10* was downregulated at 11 weeks of age, whereas *Nfkbib* and *Nfkbid* were upregulated at 11 weeks of age and then downregulated at 26 weeks of age. This suggests some involvement of the TWEAK/Fn14 pathway in the early stage of nephropathy.

Burkly et al. suggested that MAPK signaling induces stress and inflammatory responses, and that NF-κB signaling induces inflammation via MCP1, matrix metalloproteinase (MMP)-9, IL-6, and vascular cell adhesion molecule (VCAM)-1 [15]. In HIGA mice at 6 weeks, the upregulated expression of *Fn14* and *Mapk* might involve inflammatory responses in the pathogenesis of nephropathy. Furthermore, the upregulated expression of *Fn14* and *NF-κB* might be involved in the inflammatory progression in HIGA mice at 11 weeks. This indicates that Fn14 and some downstream signals associated with IgAN pathogenesis act at 6 and 11 weeks when proteinuria, IgA deposition, and mesangial proliferation are not yet evident in HIGA mice.

In addition, many genes involved in multiple biological pathways, such as TNF and cytokine signaling, were upregulated at 11 weeks of age. Interestingly, some genes in the immune network for IgA production were upregulated at 11 weeks of age, while genes in the complement pathway were upregulated at 26 weeks, indicating an advanced phase of the disease. The direct association between Fn14 and the immune network for IgA production could not be determined in the present study.

In pediatric IgA nephropathy, mesangial proliferative lesions are mainly seen, and tubular lesions are rare because of the early stage. Therefore, we analyzed Fn14 expression in the glomeruli using animal models. This is the first study to report that Fn14 is associated with early-stage glomerular lesions of IgAN in pediatric patient samples and animal models. In our study, Fn14 was not markedly involved in the pathogenesis of advanced stages of IgAN in both patient samples and animal models. Therefore, activation of the Fn14 pathway might only be a key step in the early inflammatory process, promoting inflammation by cytokine production, although the direct principal downstream targets of Fn14 at the early stage have not been identified. Furthermore, other signals might be necessary for the progression of glomerular lesions in IgAN.

There were, however, limitations to our study. First, the mechanisms leading to IgAN may differ between mouse models and humans. Second, this was a short-term study. Third, we did not perform transcriptome analysis of patient biopsy samples; therefore, we could not confirm the pronounced role of Fn14 in early lesions of IgA nephropathy in humans. In the future, it will be necessary to perform transcriptome analysis of newly onset patient biopsy samples. In addition, we did not evaluate the effect of inhibiting the TWEAK/Fn14 system as increased Fn14 expression was enhanced in the mild glomerular lesions and intact-looking tufts in IgAN patients, who were treated and achieved complete remission. We could not confirm whether the progression of IgAN was suppressed by inhibiting the Fn14 pathway. With other glomerular diseases, such as lupus nephritis, therapies that block the TWEAK/Fn14 axis, using anti-TWEAK neutralizing antibodies, have been reported to be effective [38].

We demonstrated that the Fn14 system is involved the early lesion of IgAN, therefore inhibition of the TWEAK/Fn14 pathway may be used to suppress the progression of IgAN in patients with early-stage disease when Fn14 expression increases.

## Acknowledgments

The authors would like to thank Yuko Kajita, Tokiko Mizushiro, Chihiro Tanaka, and Mio Ishimaru (Department of Pediatrics, Ehime University Graduate School of Medicine) for their technical assistance. We are thankful to Naohito Tokunaga, Takeshi Kiyoi, and Isuzu Ikeuchi of the Advanced Research Support Center of Ehime University for assistance with histological analysis of mouse kidney tissues and transcriptome analysis. We would like to thank Editage (www.editage.com) for English language editing.

## Author Contributions

**Conceptualization:** Yuko Tezuka, Minenori Eguchi-Ishimae.

**Data curation:** Yuko Tezuka, Minenori Eguchi-Ishimae, Erina Ozaki.

**Formal analysis:** Yuko Tezuka, Minenori Eguchi-Ishimae, Erina Ozaki, Toshiyuki Ito.

**Investigation:** Yuko Tezuka, Minenori Eguchi-Ishimae, Erina Ozaki, Toshiyuki Ito.

**Methodology:** Yuko Tezuka, Minenori Eguchi-Ishimae, Erina Ozaki, Mariko Eguchi.

**Project administration:** Eiichi Ishii, Mariko Eguchi.

**Software:** Yuko Tezuka, Minenori Eguchi-Ishimae, Erina Ozaki.

**Supervision:** Mariko Eguchi.

**Validation:** Eiichi Ishii, Mariko Eguchi.

**Visualization:** Yuko Tezuka, Minenori Eguchi-Ishimae.

**Writing – original draft:** Yuko Tezuka.

**Writing – review & editing:** Minenori Eguchi-Ishimae, Eiichi Ishii, Mariko Eguchi.

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
