## [Decision Letter · Decision Letter 0]

29 Mar 2021

PONE-D-20-39482

Activation of fibroblast growth factor-inducible 14 in the early phase of childhood IgA nephropathy

PLOS ONE

Dear Dr. Tezuka,

Thank you for submitting your manuscript to PLOS ONE. After careful consideration, we feel that it has merit but does not fully meet PLOS ONE’s publication criteria as it currently stands. Therefore, we invite you to submit a revised version of the manuscript that addresses the points raised during the review process.

The sequencing analysis of human biopsies is not sufficient as reported. A complete transcriptome analysis is needed to show differences between the analyzed sample groups. It would add much value to the study.

Moreover, also statistical methods and quality control for RNAseq analysis are poorly described and they need to be much more thorough.

Moreover, as explained in PLOS’s Data Policy, data set must be available at the time of publication. Sequencing Data must be deposited in a public repository and made accessible by everyone by a link or a dataset accession number that must be reported in the methods.

We look forward to receiving your revised manuscript.

Kind regards,

Fabio Sallustio, PhD

Academic Editor

PLOS ONE

Journal Requirements:

Additional Editor Comments:

EDITOR:

The sequencing analysis of human biopsies is not sufficient as reported. A complete transcriptome analysis is needed to show differences between the analyzed sample groups. It would add much value to the study.

Moreover, also statistical methods and quality control for RNAseq analysis are poorly described and they need to be much more thorough.

Moreover, as explained in PLOS’s Data Policy, data set must be available at the time of publication. Sequencing Data must be deposited in a public repository and made accessible by everyone by a link or a dataset accession number that must be reported in the methods.

Reviewers' comments:

Reviewer's Responses to Questions

**Comments to the Author**

1. Is the manuscript technically sound, and do the data support the conclusions?

Reviewer #1: Partly

2. Has the statistical analysis been performed appropriately and rigorously? 

Reviewer #1: Yes

3. Have the authors made all data underlying the findings in their manuscript fully available?

Reviewer #1: Yes

4. Is the manuscript presented in an intelligible fashion and written in standard English?

Reviewer #1: No

5. Review Comments to the Author

Reviewer #1: In this manuscript, authors described the role of TWEAK/Fn14 in pediatric IgAN.

Independently of its potential impact, the story is predominantly phenomenological/preliminary at its current stage and needs to be improved.

Major points:

-Authors should clarify the different time points in the animal model and the precise number of mice for each group.

- Authors should explain the samples used for any experimental procedures.

- IgAN patients are classified as Mild and Sever. I suggest reporting MESTC Oxford classification.

- a crucial point is the potential influence of therapy in the expression of FN14 in human biopsies. In table 1, authors reported all the therapies for each subject. However, it is necessary to specify if patients underwent therapies at the time of kidney biopsies, as this point represents a central issue in the interpretation of results and conclusion. Moreover, in figure 2 the representation of Fn14 staining in glomerular is unclear. A quantification of the intensity of Fn14 staining is necessary,

- the quality of images in figure 3 should be improved. IgA staining presents a high background and is too dark. In panel B, all the nuclei of tubular cells seem to be positive for FN14 signal, also in the control. Authors should discuss these results and should clarify the reason why they analyzed Fn14 expression only in glomeruli.

- The RNAseq results should be validated using Real Time PCR and/or tissue staining

- Authors should clarify statistic methods for RNAseq results

Minor point:

-please, improve the readability of Table 1

- the whole text needs language revisions

6. PLOS authors have the option to publish the peer review history of their article (what does this mean?). If published, this will include your full peer review and any attached files.

Reviewer #1: No

Reviewer #2

I have attached a few comments to potentially make this manuscript easier to read. This idea could have value, in that it looks at novel and potential underlying mediators of renal fibrosis that exist in IgAN, while clinically (by customary biomarkers) patients remain "stable".

While checking references of this work, in my opinion, the abstract and the introduction and through the manuscript, their statements need to be reformatted by specifically citing what evidence is referenced from animal models and what is cited from human models. On both models, it needs to be specific of what specific kidney cell they are referencing, typical examples are found on lines 74, 75,79,81.

More specific comments:

Line 58, what % of IgAN patients progress to ESRD and what are the known clinical predictive factors for progression?

Lines 96-98 seem to be repetition of statements on lines 92-94.

Line 113, "high proteinuria" is a non-acceptable term in pediatrics. Acceptable descriptors are nephrotic range (> 1.0 mg/mg in 1st am urine specimen) or non-nephrotic range (0.2 to 1.0 mg/mg).

Line 252, missing commentary addressing type and dose of treatment(s) prior to time of biopsy

Lines 292 to 298, difficult to locate on figures.

Line 385. " early stage of IgAN" adds in children. I would suggest to group the limitations of the study. Can't help to wonder why the authors did not do transcriptome analysis of human biopsies. The association of Fn14 deposition early in childhood IgAN biopsies remains loose otherwise. In terms of translational science, it warrants more robust findings and associations. This is purely descriptive, there are no statistical comparisons.

Table 1. There is no specification on time to biopsy, duration of treatment and time of follow up.

---

## [Author Response · Author response to Decision Letter 0]

23 Jun 2021

Response to the comments by Academic Editor:

Thank you for your helpful comments to our manuscript. Our answers to each of your comment are written below.

[1] The sequencing analysis of human biopsies is not sufficient as reported. A complete transcriptome analysis is needed to show differences between the analyzed sample groups. It would add much value to the study.

Reply:

We agree with your comment and a usefulness of transcriptome analysis of human biopsied samples in this study. However, because only a small number of preserved FFPE tissue samples of the patients were available for the study, we could not perform transcriptome analysis of human biopsy samples. In the future, however, we would like to proceed with transcriptome analysis for newly diagnosed cases. We have acknowledged this limitation in discussion (Page 29, Lines 447-449; Page 33 Lines 517-520)

[2] Moreover, also statistical methods and quality control for RNAseq analysis are poorly described and they need to be much more thorough.

Reply: We have added a description of statistical methods and quality control for RNAseq analysis in materials and methods (Page 19, Lines 263-265; Page 19, Lines 274-278).

[3] Moreover, as explained in PLOS’s Data Policy, data set must be available at the time of publication. Sequencing Data must be deposited in a public repository and made accessible by everyone by a link or a dataset accession number that must be reported in the methods.

Reply: We are now taking a step to register our data to DDBJ Sequence Read Archive (DRA) and will be available by public soon. A dataset accession no. will be added in the manuscript as soon as we get the number.

We have removed the phrase including “data not shown” to meet the requirement of your journal.

 

Response to the comments by Reviewer #1:

Thank you very much for your helpful comments. We have answered each of your points below.

<Major points>

[1] Authors should clarify the different time points in the animal model and the precise number of mice for each group.

Reply: In this study, three HIGA mice and 2 control mice were sacrificed at 9,11,14,26, or 40 weeks of age to obtain tissue specimens. We used 4 HIGA mice and 3 control mice in the measurement of urinary protein excretion in each week. We have added sentences in the text (Page 14, Line 185; Page 15, Lines 204-205) and Figure2.

[2] Authors should explain the samples used for any experimental procedures.

Reply: We have explained the animal models in the materials and methods sections (Page 13, Lines 164-180).

[3] IgAN patients are classified as Mild and Sever. I suggest reporting MESTC Oxford classification.

　

Reply: Thank you for your valuable advice. In this study, we treated patients based on “Guidelines for the Treatment of Childhood IgAN” proposed by the Japanese Society for Pediatric Nephrology. The patients were divided into mild and severe in this guideline. We agree the Oxford classification is more appropriate, therefore, we added MESTC Oxford classification in the text (Page 9, Lines 135-137) and Table 1. 

[4] a crucial point is the potential influence of therapy in the expression of FN14 in human biopsies. In table 1, authors reported all the therapies for each subject. However, it is necessary to specify if patients underwent therapies at the time of kidney biopsies, as this point represents a central issue in the interpretation of results and conclusion. 

Reply: Thank you for your pivotal criticism. No case of steroids or immunosuppressants were used prior to time of renal biopsy. We have added the explanation about the treatment prior to time of biopsy in the paragraph of patient samples (Page 8, Lines 126-127) and discussion (Page 29, Lines 442-444), and Table 1. We have added the details of case 14 (Table 1). 

[5] Moreover, in figure 2 the representation of Fn14 staining in glomerular is unclear. A quantification of the intensity of Fn14 staining is necessary,

 　

Reply: Thank you for your suggestions. We evaluated the intensity of Fn14 staining in the glomeruli by measuring the proportion of Fn14-stained area in each glomerulus using the ImageJ software and average values were compared at different time points (Page 17, Lines 236-240). The results are shown in Figure 3C (Page 24, Lines 360-365; Page 25, Lines 376-378).

[6] the quality of images in figure 3 should be improved. IgA staining presents a high background and is too dark. In panel B, all the nuclei of tubular cells seem to be positive for FN14 signal, also in the control. Authors should discuss these results and should clarify the reason why they analyzed Fn14 expression only in glomeruli.

Reply: We have improved the quality of images in figure 3. We have revised the term (Page 3, Lines 38) and added terms to clarify the sentences (Page 30, Line 466, 468).

According to THE HUMAN PROTEIN ATLAS [https://www.proteinatlas.org/], Fn14 staining by antibody is not detected on cells in glomeruli, but moderate staining is detectable on cells in tubules. Regarding the comment “all the nuclei of tubular cells seem to be positive for FN14”, we think these staining is reflecting the physiological staining of Fn14 in the tubules. In pediatric IgA nephropathy, mesangial proliferation in glomeruli is frequently observed findings compared to tubular lesion which is rarely seen because of early stage. Therefore, we analyzed Fn14 expression only in glomeruli in this study.

We added sentences in the paragraph of discussion (Page 32, Lines 505-507).

[7] The RNAseq results should be validated using Real Time PCR and/or tissue staining

Reply: Thank you for your suggestion. We tried the validation of RNAseq results by real-time PCR, but sufficient RNA samples for reliable validation with real-time PCR were not available after RNAseq. 

[8] Authors should clarify statistic methods for RNAseq results

 Reply: We have described statistic methods for RNAseq results in the paragraph of Materials and Methods (Page 19, Line 274-278)

 

<Minor point>

[9] Please, improve the readability of Table 1

 Reply: We have brushed up the items listed in the table to make it readable.

[10] the whole text needs language revisions

 Reply: We have reviewed and corrected the text with the help of English editing service (Page 34, Line 536-537).

 

Response to the comments by Reviewer #2:

Thank you very much for your comments. Our answers to your points are as follows.

The reference number has been changed when adding some sentences in revised manuscript. We have also added or revised the text to clarify.

[1] While checking references of this work, in my opinion, the abstract and the introduction and through the manuscript, their statements need to be reformatted by specifically citing what evidence is referenced from animal models and what is cited from human models. On both models, it needs to be specific of what specific kidney cell they are referencing, typical examples are found on lines 74, 75,79,81.

 Reply: We extensively revised and clarified what specific kidney cell referencing on both models in introduction (Page 6, Lines 87-88, 90-91), by describing the specific animal models and human diseases reported (Page 6, Lines 92-93; Page 6-7, Lines 94-98).

<More specific comments>

[2] Line 58, what % of IgAN patients progress to ESRD and what are the known clinical predictive factors for progression?

Reply: We have added sentence about the prognosis and clinical and histological predictive factors in the introduction (Page 5, Lines 62-72), mainly about the following points; 

Among pediatric patients with IgA nephropathy in Japan, 11% have been reported to develop ESRD within 15 years. 

Clinical predictors of a poor outcome were a low GFR, a high mean blood pressure and a high amount of albuminuria at time of biopsy, and low GFR and a high albuminuria during follow up. 

Histological predictors were mesangial hypercellularity, endocapillary hypercellularity, tubular atrophy and crescents. 

While the incidence of pediatric IgA nephropathy patients who show hypertension or decreased renal function at onset is relatively rare in comparison with adults, proteinuria is the most important risk factor for progression of the disease in childhood, in particular, the degree of proteinuria during the follow-up period.

[3] Lines 96-98 seem to be repetition of statements on lines 92-94.

Reply: We have eliminated the sentence corresponds to lines 96-98 (Page 7, Lines 112-114 in revised, markup manuscript).

[4] Line 113, "high proteinuria" is a non-acceptable term in pediatrics. Acceptable descriptors are nephrotic range (> 1.0 mg/mg in 1st am urine specimen) or non-nephrotic range (0.2 to 1.0 mg/mg).

Reply: As your suggestion, the term “high proteinuria” is an inappropriate description. In this study, we treated based on “Guidelines for the Treatment of Childhood IgAN” proposed by the Japanese Society for Pediatric Nephrology. In this guideline, amount of urinary protein is stratified by whether early morning urinary protein / creatinine ratio exceeds 1.0 or not, therefore, we have replaced the term “high proteinuria” throughout the paper with “heavy proteinuria” to use more precise term (Page 8, Lines 130). While nephrotic range refers 2.0 or higher of urinary protein / creatinine ratio, we did not use the term “nephrotic range” in our study. 

[5] Line 252, missing commentary addressing type and dose of treatment(s) prior to time of biopsy.

Reply: We have added explanation about the treatment prior to time of biopsy in the paragraph of patient samples (Page 8, Lines 126-127) and Table 1. 

We also have mentioned about the treatment in the discussion (Page, 29, Lines 442-444).

[6] Lines 292 to 298, difficult to locate on figures.

 Reply: Thank you for your suggestion. We show the ratio of the mean U-pro / Cr in HIGA mice (n = 4) to the mean U-pro/Cr in control mice (n = 3), as 1 at 9 weeks. We have added the annotations of the vertical axis of the graph (Figure 2 B) to “U-pro/Cr, the ratio of HIGA mice to control”. We also have added the text in paragraph of results (Page 22, Lines 317).

Also, since the proliferation of the mesangial matrix was more noticeable than the mesangial cell proliferation, we have revised the sentence to clarify (Page 22, Lines 325-330).

[7] Line 385. "early stage of IgAN" adds in children. I would suggest to group the limitations of the study. Can't help to wonder why the authors did not do transcriptome analysis of human biopsies. The association of Fn14 deposition early in childhood IgAN biopsies remains loose otherwise. In terms of translational science, it warrants more robust findings and associations. This is purely descriptive, there are no statistical comparisons.

Reply: Thank you for your important suggestion. We have added “IgAN in childhood” (Page 29, Lines 445). We agree with your comment and consider this point is important. In this study, we used preserved FFPE tissue samples from patients with pediatric IgAN and only small number of samples were available for the experiments. Therefore, we could not perform transcriptome analysis of human biopsy samples. This is the limitation in this study. In the future, however, we would like to proceed with transcriptome analysis for newly diagnosed cases.

We have acknowledged this limitation in discussion (Page 29, Lines 447-449; Page 33, Lines 517-520).

[8] Table 1. There is no specification on time to biopsy, duration of treatment and time of follow up.

Reply: We have revised the Table 1 including the period and treatment prior to biopsy, and follow-up period.

Thank you for giving us the opportunity to strengthen our manuscript with your valuable comments and queries.

---

## [Decision Letter · Decision Letter 1]

26 Jul 2021

PONE-D-20-39482R1

Activation of fibroblast growth factor-inducible 14 in the early phase of childhood IgA nephropathy

PLOS ONE

Dear Dr. Tezuka,

Thank you for submitting your manuscript to PLOS ONE. After careful consideration, we feel that it has merit but does not fully meet PLOS ONE’s publication criteria as it currently stands. Therefore, we invite you to submit a revised version of the manuscript that addresses the points raised during the review process.

ACADEMIC EDITOR:

Please, provide in the text of "Methods" the accession Number for your Sequencing Data in the public repository.Please, add more details and references about the mouse model of IgAN in the introduction.

We look forward to receiving your revised manuscript.

Kind regards,

Fabio Sallustio, PhD

Academic Editor

PLOS ONE

Journal Requirements:

Additional Editor Comments (if provided):

Please, provide in the text of "Methods" the accession Number for your Sequencing Data in the public repository.

Please, add more details and references about the mouse model of IgAN in the introduction.

Reviewers' comments:

Reviewer's Responses to Questions

**Comments to the Author**

1. If the authors have adequately addressed your comments raised in a previous round of review and you feel that this manuscript is now acceptable for publication, you may indicate that here to bypass the “Comments to the Author” section, enter your conflict of interest statement in the “Confidential to Editor” section, and submit your "Accept" recommendation.

Reviewer #1: All comments have been addressed

2. Is the manuscript technically sound, and do the data support the conclusions?

Reviewer #1: Yes

3. Has the statistical analysis been performed appropriately and rigorously? 

Reviewer #1: Yes

4. Have the authors made all data underlying the findings in their manuscript fully available?

Reviewer #1: Yes

5. Is the manuscript presented in an intelligible fashion and written in standard English?

Reviewer #1: Yes

6. Review Comments to the Author

Reviewer #1: (No Response)

7. PLOS authors have the option to publish the peer review history of their article (what does this mean?). If published, this will include your full peer review and any attached files.

Reviewer #1: No

---

## [Author Response · Author response to Decision Letter 1]

7 Sep 2021

Response to the comments by Academic Editor:

Thank you for your helpful comment on our manuscript. Our answers to each of your comments are provided below.

[1] Please, provide in the text of "Methods" the accession Number for your Sequencing Data in the public repository.

Reply:

1. We have mentioned the accession number for our sequencing data in the public repository in the text of “Materials and methods” (Page 20, lines 282-283).

[2] Please add more details and references about the mouse model of IgAN in the introduction. 

Reply: Thank you for your comment. We have added a description of the mouse model of IgAN in the Introduction (Page 7, lines 109-113) and have also listed the corresponding references (references 30-34). Reference list has also been updated (Page 39, lines 627-639).

---

## [Editor Report · Decision Letter 2]

20 Sep 2021

Activation of fibroblast growth factor-inducible 14 in the early phase of childhood IgA nephropathy

PONE-D-20-39482R2

Dear Dr. Tezuka,

We’re pleased to inform you that your manuscript has been judged scientifically suitable for publication and will be formally accepted for publication once it meets all outstanding technical requirements.

Kind regards,

Fabio Sallustio, PhD

Academic Editor

PLOS ONE
---

## [Editor Report · Acceptance letter]

24 Sep 2021

PONE-D-20-39482R2 

Activation of fibroblast growth factor-inducible 14 in the early phase of childhood IgA nephropathy 

Dear Dr. Tezuka:

I'm pleased to inform you that your manuscript has been deemed suitable for publication in PLOS ONE. Congratulations! Your manuscript is now with our production department. 

Kind regards, 

on behalf of

Dr. Fabio Sallustio 

Academic Editor

PLOS ONE